# Real-World Technical Hurdles of ctDNA NGS Analysis: Lessons from Clinical Implementation

**DOI:** 10.3390/diseases13100312

**Published:** 2025-09-23

**Authors:** Simon Cabello-Aguilar, Julie A. Vendrell, Jérôme Solassol

**Affiliations:** 1Laboratoire de Biologie des Tumeurs Solides, CHU Montpellier, Université de Montpellier, 34295 Montpellier, France; s-cabelloaguilar@chu-montpellier.fr (S.C.-A.); j-vendrell@chu-montpellier.fr (J.A.V.); 2Montpellier BioInformatics for Clinical Diagnosis (MoBiDiC), Plateau de Médecine Moléculaire et Génomique (PMMG), CHU Montpellier, 34295 Montpellier, France; 3INSERM U1194, Institut de Recherche en Cancérologie de Montpellier (IRCM), Université de Montpellier, 34090 Montpellier, France

**Keywords:** ctDNA, NGS, liquid biopsy, precision oncology

## Abstract

Next-generation sequencing (NGS) of circulating tumor DNA (ctDNA) represents a minimally invasive alternative to conventional tissue biopsies, providing real-time genomic snapshots of heterogeneous tumors from blood draws. This liquid biopsy approach has demonstrated significant utility for early detection, molecular profiling, and monitoring treatment response in cancer patients. However, significant barriers to widespread clinical implementation still remain, such as a lack of standardized methods for ctDNA content quantification and limited variant detection sensitivity at ultra-low frequencies. Herein, we discuss three key improvements: (i) reducing the limit of detection (LoD) from 0.5% to 0.1%, which would increase alteration detection from 50% to approximately 80%; (ii) developing a dynamic LoD approach calibrated to sequencing depth, thereby enhancing result reliability and confidence in clinical interpretation; and (iii) utilizing strategic bioinformatics pipelines with “allowed” and “blocked” lists to enhance accuracy while minimizing false positives. While ctDNA analysis remains approximately 30% less sensitive than tissue-based testing, addressing these limitations through technological advancement and standardization protocols could accelerate integration into routine clinical practice, potentially transforming cancer management while reducing healthcare costs.

## 1. Liquid Biopsy in Modern Oncology: The Dawn of a New Diagnostic Era

The advent of precision medicine that considers tumor-specific molecular characteristics to tailor therapies to each patient has recently revolutionized cancer treatment. While robust clinical evidence highlights the power of next-generation sequencing (NGS) to improve patient survival by identifying actionable alterations in tissue biopsies [1], we stand at the threshold of a promising future base for NGS: the liquid biopsy.

The elegance of this approach lies in its simplicity: a blood draw replacing an invasive procedure, while potentially capturing a comprehensive picture of heterogeneous disease. It involves the analysis of circulating cell-free DNA (cfDNA) fragments released into the bloodstream, with particular focus on tumor-derived fragments, known as circulating tumor DNA (ctDNA) [2].

Unlike conventional invasive tissue biopsies, which are often limited by tumor heterogeneity, liquid biopsies offer a minimally invasive alternative that provides a more comprehensive and real-time genomic snapshot of the tumor, capturing information from both primary and metastatic lesions. As such, they are useful for early cancer detection, molecular profiling, monitoring of minimal residual disease (MRD), assessment of treatment response, and identification of resistance mechanisms [3,4]. The clinical implementation of ctDNA NGS has demonstrated a measurable impact on oncological practice across multiple malignancies. In non-small cell lung cancer (NSCLC), ctDNA-based mutation detection has achieved guideline inclusion as a standard diagnostic modality for identifying actionable alterations in *EGFR*, *KRAS*, and *MET*, facilitating timely therapeutic intervention while circumventing the morbidity and temporal constraints associated with invasive tissue sampling procedures [5]. In metastatic colorectal carcinoma, ctDNA assays enable the detection of *KRAS* and *BRAF* mutations that serve as predictive biomarkers for anti-EGFR monoclonal antibody efficacy, with enhanced detection rates when combined with tissue-based testing [6]. For estrogen receptor-positive breast carcinoma, ctDNA surveillance can identify acquired *ESR1* mutations associated with endocrine therapy resistance, with FDA approval of the Guardant360 CDx test specifically for detecting *ESR1* mutations to guide elacestrant treatment decisions [7]. Importantly, ctDNA-based assays also enable longitudinal therapy monitoring by detecting emerging resistance mutations in real time, thereby guiding timely treatment adaptation. A classic example is the detection of the *EGFR* T790M resistance mutation in NSCLC patients treated with first- or second-generation EGFR inhibitors. The appearance of this mutation in the blood often precedes clinical or radiographic progression, allowing clinicians to proactively switch to a third-generation inhibitor like osimertinib.

Tissue-based analysis currently remains the gold standard and first-line approach in clinical practice [8], with ctDNA analysis employed as a complementary tool in cases of therapeutic urgency, non-contributory tissue results, or non-feasible tissue sampling.

Among the available technologies for ctDNA analysis, digital droplet PCR (ddPCR) offers high sensitivity for detecting specific mutations [9] but has low throughput and is unable to detect a wide range of mutations simultaneously. NGS is capable of identifying a broad spectrum of genetic alterations, including point mutations, copy number variations, and gene translocations, and could thus provide a more powerful alternative [10].

In lung cancer, prospective studies have shown that ctDNA NGS can identify actionable mutations when tissue is limited and predict recurrence risk and adjuvant chemotherapy benefit after surgery [11,12]. In colorectal cancer, the ORCA trial is providing early evidence that longitudinal ctDNA monitoring during systemic therapy enables dynamic assessment of treatment response and may support early intervention upon molecular progression. Beyond tumor-specific settings, ctDNA has also been applied to optimize patient selection in early-phase clinical trials, supporting more efficient enrollment in precision oncology studies [13], and decentralized genomic profiling across oncology centers, illustrating the feasibility of large-scale clinical implementation. These examples reinforce how ctDNA NGS is transitioning from investigational use toward integration into clinical practice.

Though the clinical utility of ctDNA NGS requires further validation, its time-saving capacity cannot be understated, particularly concerning aggressive cancers where rapid identification of actionable mutations may enable prompt initiation of targeted therapies, potentially improving survival outcomes [14,15]. Timely detection and intervention are essential for effective cancer management, not only from a clinical perspective but also from an economic and societal standpoint [16]. Indeed, early intervention can reduce the need for more intensive and prolonged treatments, leading to substantial cost savings for healthcare systems and better outcomes for patients.

With such obvious advantages, why is ctDNA NGS not yet practiced in every hospital laboratory worldwide? Despite the considerable potential of NGS-based ctDNA analysis, several technical and practical challenges unfortunately hinder its widespread adoption into clinical practice. Herein, we identify the key limitations and propose methodological improvements that could accelerate the integration of this cutting-edge technology into standard oncological care.

## 2. Variant Detection and Coverage: Navigating a Complex Relationship

The low abundance of tumor-derived DNA against a large background of normal DNA presents a significant challenge in ctDNA analysis. Indeed, the detection of clinically relevant alterations is exceptionally difficult due to variant allele frequencies (VAFs) for such alterations frequently falling below 1% at early disease stages or after curative-intent treatment. Methods with sufficient sensitivity to detect variants at ultralow frequencies (commonly below 1% and as low as 0.05% in clinical practice) are required to overcome these challenges.

For a variant to be considered as true, it must be supported by at least *n* individual reads, with the value of *n* set high enough to avoid the reporting of false variants due to sequencing errors, yet not too high to avoid missing true variants. While *n* = 5 works well with DNA extracted from formalin-fixed paraffin-embedded (FFPE) tissue samples, it should be lowered to *n* = 3 to achieve the sensitivity needed for ctDNA analysis. This is feasible since the DNA in liquid biopsies is not prone to cytosine deamination [17]. The probability of detecting a variant supported by at least three unique reads can be modeled as a function of the depth of coverage (DoC) using a binomial law, where the probability of successful detection is equal to the VAF [18]. For six different VAFs from 0.1% to 1%, the DoC required for a 99% detection probability ranges from 1000× for a VAF of 1% to approximately 10,000× for a VAF of 0.1% (Figure 1A). While increasing the DoC clearly enhances detection probability, achieving a 99% detection rate for very low frequencies remains challenging and significantly raises sequencing costs. Major commercial therapy selection panels such as Guardant360 CDx or FoundationOne Liquid CDx typically achieve a raw coverage of ~15,000×, which, after deduplication, yields an effective depth of ~2000×—consistent with their reported LoD of ~0.5%. Some researchers propose ultra-deep sequencing as a solution, with recommendations of up to 20,000 unique reads per base [19]. However, implementing such high-coverage sequencing in routine clinical laboratories is simply not feasible, requiring ultra-high throughput sequencers that remain prohibitively expensive and not widely accessible.

## 3. Duplicated Reads: Avoiding Redundancy, Enhancing Sensitivity

NGS sequencer output contains numerous duplicate reads, necessitating adjustments in the subsequent bioinformatics pipelines. The most accurate method for deduplicating reads is including a unique molecular identifier (UMI) barcoding step during library preparation to allow for removal of duplicate reads during post-processing. UMIs are short sequences added to DNA fragments in certain NGS protocols to identify the original input DNA molecules. These tags are incorporated prior to PCR amplification, helping to minimize errors and reduce quantitative biases introduced during the amplification process. Their use in cfDNA NGS experiments has proven highly advantageous, allowing for better distinction between true signals and background noise [20]. In practice, the UMI deduplication yield is approximately 10% under optimal sequencing conditions (including DNA quality, DNA quantity, and library preparation). Variant calling is performed on this much-reduced fraction of deduplicated reads, an important consideration when calculating the number of samples to multiplex in a run, based on panel size. For example, a DoC of 20,000× before deduplication would result in approximately 2000× after deduplication, which is insufficient for ultra-low variant detection.

Also noteworthy is that while essential, UMI-based deduplication is technically challenging with no universally accepted methodology and thus can only be performed by skilled bioinformaticians.

## 4. DNA Quantity Drives Variant Discovery

Unlike the often damaged DNA that is extracted from FFPE tissue samples, cfDNA is typically good quality and naturally fragmented into lengths appropriate for library preparation with short-read sequencers. A critical limiting factor for ctDNA NGS assay accuracy, however, is the amount of available input DNA. The number of DNA strands available for library preparation depends on the input DNA mass, with 1 ng of human genomic DNA corresponding to approximately 300 haploid genome equivalents (GEs) [21]. Achieving 20,000× coverage after deduplication requires a minimum input of 60 ng DNA. The quantity of cfDNA in cancer patients is highly variable and influenced by factors such as tumor type, stage, and volume. While cancer patients generally exhibit higher cfDNA levels than healthy individuals, this pattern differs across histologies. For instance, lung cancers can have low cfDNA levels (5.23 ± 6.4 ng/mL), while liver cancers often show much higher levels (46.0 ± 35.6 ng/mL) [22]. This is critically important because the ultimate constraint on sensitivity is the absolute number of mutant DNA fragments in a sample. For example, a 10 mL blood draw from a lung cancer patient might yield only ~8000 haploid genome equivalents (GEs). If the ctDNA fraction is 0.1%, this provides a mere eight mutant GEs for the entire analysis, making detection statistically improbable. Conversely, the same volume from a high-shedding liver cancer patient could provide ~80,000 GEs. Even with the same 0.1% ctDNA fraction, this yields 80 mutant GEs, providing a much stronger signal that is more amenable to detection by means of ultra-deep sequencing. Therefore, while a high background of normal DNA presents challenges, the higher total DNA quantity from high-shedding tumors is often advantageous because it is typically associated with a greater absolute number of targetable tumor molecules.

## 5. Quantifying the ctDNA Fraction: An Ongoing Challenge

Accurate quantification of the ctDNA fraction within cfDNA is crucial for generating reliable results. However, the absence of standardized methods for assessing ctDNA content currently hinders the broader clinical implementation of ctDNA NGS analyses. Among the available approaches, aneuploidy measurements obtained from low to ultra-low pass sequencing [23] lack sufficient accuracy when ctDNA levels fall below 5%, thereby highlighting their insufficient sensitivity for most ctDNA analyses. An alternative method relying on the maximum VAF of non-germline genetic alterations [24] fails to account for tumor heterogeneity and is highly dependent on gene panel design. Broader panels increase the likelihood of detecting variants shared by all tumor cells in the sample, thus more accurately reflecting the true ctDNA fraction. While promising technologies such as fragmentomics and methylomics are emerging to potentially overcome these limitations, approaches robust enough for implementation in routine clinical practice are required.

## 6. Lowering the LoD: An Essential Step to Enhance Clinical Utility

One key parameter in ctDNA NGS is the limit of detection (LoD), as it determines the minimum VAF at which a variant can be reliably detected. Most commercially available ctDNA NGS assays report an LoD of 0.5% [25]. We sought to evaluate the clinical impact of this LoD threshold in practice by examining results from over 2500 samples processed using droplet digital PCR (ddPCR) in our laboratory. Remarkably, among the positive cases, only half of the detected alterations exhibited a VAF above 0.5% (Figure 1B), highlighting the insufficient sensitivity of conventional NGS approaches compared to ddPCR. The detection sensitivity can be substantially increased by lowering the LoD to 0.1%, allowing the successful identification of approximately 80% of alterations (Figure 1B). The resulting improvement in analytical sensitivity by lowering the LoD to 0.1% would significantly enhance the clinical utility of ctDNA testing, potentially transforming patient management across various oncological settings.

An important consideration is, however, the increased incidence of false positives arising from technical artifacts or sequencing errors that may occur by lowering the LoD. To confidently allow such a low LoD would therefore require well-optimized sequencing experiments and sufficient input DNA quantities.

## 7. From Limitations to Confidence: Interpreting Negative Results in ctDNA Analysis

Multiple limitations currently impede the accurate analysis of ctDNA using NGS, particularly the confident reporting of negative results. We present a novel analytical approach that enhances confidence in calling negative ctDNA results. By leveraging the theoretical LoD as a function of sequencing depth (Figure 1A), we may transform a previously considered limitation into an analytical advantage. The minimum VAF detectable with high confidence can be precisely calculated for each sample based on its achieved coverage metrics. For instance, a VAF of 0.2% can be detected at a probability exceeding 99% with a DoC of 5000×. Consequently, should no genetic alteration be detected at this depth, one can confidently conclude upon the absence of variants with an LoD of 0.2% for this specific sample.

To effectively mitigate the abovementioned risks associated with lowering the LoD, sophisticated strategies can be implemented within the bioinformatics pipelines. A particularly powerful approach involves utilizing “allowed list” and “blocked list” annotation steps. The “allowed list” encompasses variants with established diagnostic, prognostic, or theranostic relevance, while the “blocked list” flags variants known to be artifacts. These artifacts are recurrent false positives arising from non-biological sources, and filtering them is crucial for achieving a low LoD. Common examples include the following: (i) systematic base-calling errors in specific sequence contexts (e.g., GGC motifs); (ii) reads misaligned from homologous pseudogenes; (iii) oxidative damage-induced mutations (e.g., G>T transversions); and (iv) true somatic variants originating from clonal hematopoiesis (CHIP) in blood cells, which are not representative of the tumor’s genome. This systematic methodology significantly minimizes the impact of sequencing errors and substantially improves variant calling accuracy, ultimately enhancing clinical decision-making.

## 8. Conclusions: Unleashing the Full Potential of Cancer Care

Whilst standing at the frontier of a revolution in cancer care, we must endure unavoidable growing pains. While the emergence of NGS-based ctDNA analysis in everyday clinical practice justifiably inspires much enthusiasm, substantial limitations remain that require our collective attention to finely tune this powerful tool.

In current practice, ctDNA analysis offers a crucial window into the molecular landscape when tissue sampling is elusive, as is often the case with hard-to-access tumors. A simple blood draw replaces an invasive procedure and captures a comprehensive picture of the heterogeneous disease. Perhaps most compelling is the acceleration of clinical decision-making. In aggressive malignancies, weeks saved in treatment initiation can extend patient lives and effectively reduce healthcare costs. Herein, we have proposed a dynamic LoD approach allowing for the adaptation of sensitivity thresholds based on sequencing depth, which we believe provides further support for the eventual large-scale implementation of ctDNA analysis in clinical laboratories.

It is important to acknowledge that ctDNA sensitivity remains approximately 30% below tissue-based analysis, which rightfully maintains its gold standard status [26]. Furthermore, the technical challenges facing widespread adoption of ctDNA remain substantial: insufficient DNA input quantities from patients with low cfDNA, coverage limitations with current sequencing technologies, and complex bioinformatics requirements. Figure 2 summarizes the main technical hurdles and conceptual advances, offering a visual roadmap for overcoming barriers to clinical adoption. The route towards routine clinical integration of ctDNA has now been laid, with three compulsory pit stops to address outstanding key issues. These are as follows: reducing the LoD to values that better reflect real-world cancer samples (to about 0.1%); establishing internationally standardized protocols for UMI deduplication; and, critically, developing reliable methods for accurate quantification of the ctDNA fraction within total cfDNA.

While we remain convinced that liquid biopsy represents the future of precision oncology, we invite laboratories considering the adoption of ctDNA NGS to exercise cautious enthusiasm. Only with a collective awareness of the need for rigorous validation, standardization, and further technological advancement can we unleash the full potential of ctDNA NGS with widespread implementation.

## Figures and Tables

**Figure 1 diseases-13-00312-f001:**
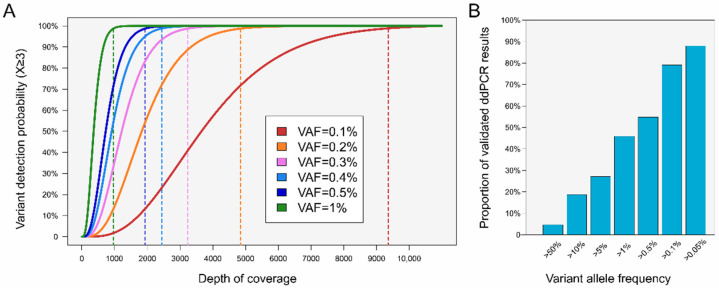
Impact of depth of coverage (DoC) on variant detection probability according to variant allele frequency (VAF) in routine ctDNA analysis. (**A**). Detection probability of a genetic alteration supported by at least three unique reads, plotted as a function of DoC for six different VAFs ranging from 0.1% to 1%. Vertical dashed lines indicate the DoC required to achieve a 99% detection probability. (**B**). Proportion of positive results as detected by means of droplet digital PCR (ddPCR) in our laboratory (*n* = 682), categorized according to their VAF.

**Figure 2 diseases-13-00312-f002:**
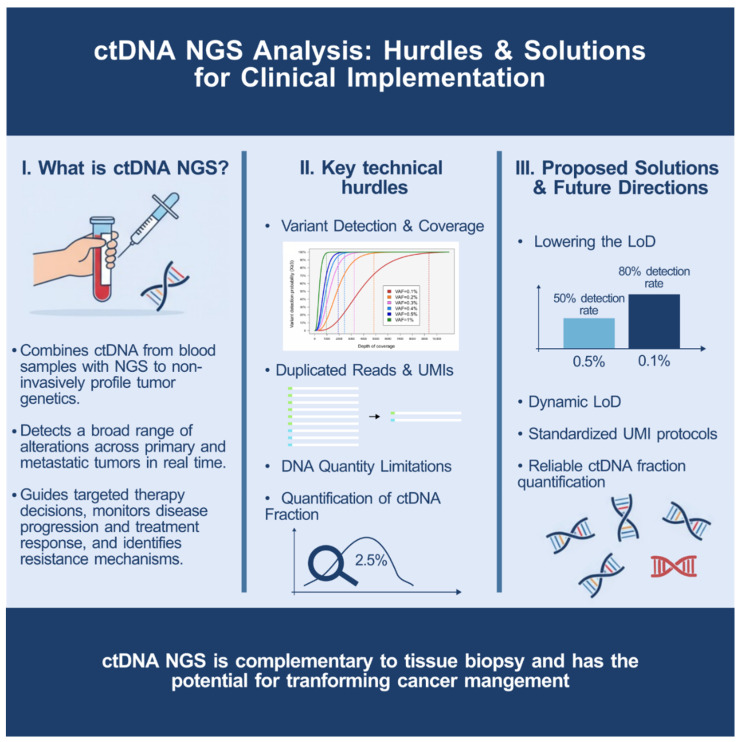
Overview of ctDNA NGS analysis hurdles and solutions.

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
