# Peer review of "Real-World Technical Hurdles of ctDNA NGS Analysis: Lessons from Clinical Implementation"

_diseases, 2025, doi:10.3390/diseases13100312_

Round 1

Reviewer 1 Report

Comments and Suggestions for Authors

Aguilar et al. have written a timely commentary on the real-world challenges of implementing ctDNA next-gen sequencing, liquid biopsy in clinical practice. 

ctDNA approach provides a minimally invasive alternative to the traditional tissue biopsies, providing real time genomic snapshots of tumors which are often heterogeneous. 

The authors have expanded on the key hurdles: depth of coverage requirements, input limitations, and limit of detection, and have highlighted the need to improve analytical sensitivity and interpretation.

There are several areas that need improvement. Below are my observations:

While the technical aspect of the commentary is strong, it would benefit from a stronger clinical contextualizing/framing. For example:

  • Specific cancer lineages where ctDNA NGS has shown the greatest promise and will continue to develop in the coming decade.
  • Examples of clinical trials or regulatory challenges.
  • Implications for therapy monitoring (such as resistance, refractory cancers).
  • Comparison with emerging technologies (e.g., nanopore sequencing, hybrid capture panels) would provide better context within the current landscape of clinical genomics.
  • Consider adding an informative figures that provide a comprehensive overview of all the key points which also help readers consolidate their understanding.

Author Response

Reviewer 1 :

Aguilar et al. have written a timely commentary on the real-world challenges of implementing ctDNA next-gen sequencing, liquid biopsy in clinical practice. 

ctDNA approach provides a minimally invasive alternative to the traditional tissue biopsies, providing real time genomic snapshots of tumors which are often heterogeneous. 

The authors have expanded on the key hurdles: depth of coverage requirements, input limitations, and limit of detection, and have highlighted the need to improve analytical sensitivity and interpretation.

There are several areas that need improvement. Below are my observations:

While the technical aspect of the commentary is strong, it would benefit from a stronger clinical contextualizing/framing.

Thank you for your insightful feedback. We fully agree that strengthening the clinical context will significantly enhances the impact of our commentary. To address your comments, we revised the manuscript to integrate concrete clinical applications, trial data, therapy monitoring examples, and emerging technologies. The main changes are summarized below, with indications of where they appear in the manuscript.

  1. Reviewer comment: Specific cancer lineages where ctDNA NGS has shown the greatest promise and will continue to develop in the coming decade.

We agree that grounding the discussion in clinical practice improves readability and impact. Accordingly, in the section “Liquid Biopsy in Modern Oncology: The Dawn of a New Diagnostic Era” (p. 3, l. 17), we added examples from major tumor types:

“The clinical implementation of ctDNA NGS has demonstrated measurable impact on oncological practice across multiple malignancies. In non-small cell lung cancer (NSCLC), ctDNA-based mutation detection has achieved guideline inclusion as a standard diagnostic modality for identifying actionable alterations in EGFR, KRAS, and MET, facilitating timely therapeutic intervention while circumventing the morbidity and temporal constraints associated with invasive tissue sampling
[4] In metastatic colorectal carcinoma, ctDNA assays enable detection of KRAS and BRAF mutations that serve as predictive biomarkers for anti-EGFR therapy efficacy, with enhanced detection rates when combined with tissue-based testing [5].”

  1. Reviewer comment :Examples of clinical trials or regulatory challenges.

We agree that prospective trials provide essential real-world framing. To address this, we added new examples in the introduction (“Liquid Biopsy in Modern Oncology”, p. 4, l. 18):

“In lung cancer, prospective studies have shown that ctDNA NGS can identify actionable mutations when tissue is limited and can predict recurrence risk and adjuvant chemotherapy benefit after surgery [10, 11]. In colorectal cancer, the ORCA trial is providing early evidence that longitudinal ctDNA monitoring during systemic therapy enables dynamic assessment of treatment response and may support early intervention upon molecular progression. Beyond tumor-specific settings, ctDNA has also been applied to optimize patient selection in early-phase clinical trials, supporting more efficient enrollment in precision oncology studies [12], and to decentralized genomic profiling across oncology centers, illustrating the feasibility of large-scale clinical implementation. These examples reinforce how ctDNA NGS is transitioning from investigational use toward integration into clinical practice.”

  1. Reviewer comment : Implications for therapy monitoring (such as resistance, refractory cancers).

The original version briefly mentioned therapy monitoring but we agree with the reviewer that it lacks specific examples. We strengthened this aspect in “Liquid Biopsy in Modern Oncology” (p. 3, l. 25) by adding:

“For estrogen receptor-positive breast carcinoma, ctDNA surveillance can identify acquired ESR1 mutations associated with endocrine therapy resistance, with FDA approval of the Guardant360 CDx test specifically for detecting ESR1 mutations to guide elacestrant treatment decisions [6]. Importantly, ctDNA-based assays also enable longitudinal therapy monitoring by detecting emerging resistance mutations in real time, thereby guiding timely treatment adaptation. A classic example is the detection of the EGFR T790M resistance mutation in NSCLC patients treated with first- or second-generation EGFR inhibitors. The appearance of this mutation in the blood often precedes clinical or radiographic progression, allowing clinicians to proactively switch to a third-generation inhibitor like osimertinib.”

  1. Reviewer comment: Comparison with emerging technologies (e.g., nanopore sequencing, hybrid capture panels) would provide better context within the current landscape of clinical genomics.

We thank the reviewer for this insightful suggestion regarding the technological context. The original version primarily contrasted ddPCR with NGS; however, we agree that acknowledging additional NGS approaches and emerging technologies provides a more comprehensive overview. Accordingly, we have added a new paragraph (p. 9, l. 14) to broaden this discussion.

Technological Strategies: From Amplicons to Long Reads

Contemporary ctDNA NGS implementation requires consideration of library preparation methodologies and emerging technological platforms. Amplicon-based approaches demonstrate superior turnaround time (1 day versus 3 days) and reduced DNA input requirements compared to hybrid-capture methods, though hybrid-capture-based panels exhibit enhanced uniformity and comprehensive genomic coverage [24]. Long-read sequencing platforms, particularly Oxford Nanopore Technologies, enable simultaneous detection of DNA methylation patterns and fragmentomic signatures without bisulfite conversion, with demonstrated capability to identify cell-of-origin and cancer-specific methylation features from single-molecule resolution analysis [25]. However, achieving sufficient sequencing coverage (>10 000×) remains challenging with nanopore platforms for gene-panel applications, as current nanopore protocols are optimized for DNA molecules much longer than the average ctDNA fragment, limiting coverage efficiency for ctDNA detection [26].”

  1. Consider adding an informative figures that provide a comprehensive overview of all the key points which also help readers consolidate their understanding.

We appreciate the reviewer's suggestion regarding the inclusion of an informative overview figure. In response to this valuable feedback, we have incorporated Figure 2, which provides a comprehensive visual summary of the key technical challenges and emerging solutions in ctDNA analysis. This figure is introduced in our conclusion with the following statement:

“Figure 2 summarizes the principal technical challenges and conceptual advances, offering a visual roadmap for overcoming barriers to clinical adoption.”

Reviewer 2 Report

Comments and Suggestions for Authors

This is an excellent review! The presentation is very concise, clear and well-structured.

I have just a few minor questions.

The authors quote the suggestion to increase sequencing depth up to 20,000 reads per base in order to achieve proper sensitivity of NGS ctDNA testing. Did someone validate this statement? The capacity of NGS facilities is rapidly growing, therefore, ultra-deep coverage should not be a problem in the near future. If 20,000 coverage becomes easily affordable, would it be a solution? What is the coverage of well-known commercial tests, e.g., CancerSeek?

Lines 128-139: One cannot discuss tumor categories without adjusting for tumor volume. What is the minimal size of, say, lung or liver cancer lump, to be detectable by an “ideal” ctDNA assay from 10 ml of blood? I also do not understand the message of this section: if one considers cfDNA (normal and tumor DNA), why the increased concentration of cfDNA is better?! If this increased concentration is due to the excess of normal DNA, formal VAF value of 0.1% is no longer applicable, so even higher sensitivity of ctDNA detection is required. Please check this section.   

What are “variants known to be artifacts” in the bioinformatic “blocked list”? – please explain.

Author Response

Reviewer 2 :

This is an excellent review! The presentation is very concise, clear and well-structured.

I have just a few minor questions.

We sincerely thank the reviewer for their positive feedback and for raising these important questions. We agree that clarifying these points will significantly strengthen our commentary. Below, we address each question in detail, with references to where modifications were made in the manuscript.

  1. The authors quote the suggestion to increase sequencing depth up to 20,000 reads per base in order to achieve proper sensitivity of NGS ctDNA testing. Did someone validate this statement? The capacity of NGS facilities is rapidly growing, therefore, ultra-deep coverage should not be a problem in the near future. If 20,000 coverage becomes easily affordable, would it be a solution? What is the coverage of well-known commercial tests, e.g., CancerSeek?

We thank the reviewer for bringing up this question. The recommendation for 20,000x coverage is less a single validated experimental mandate and more a theoretical target derived from statistical modeling needed to overcome the profound challenge of signal-to-noise at ultra-low VAFs. As our own Figure 1A illustrates, detecting a 0.1% VAF variant with 99% probability requires ~10,000x coverage after UMI deduplication. Given that UMI deduplication yields are often around 10%, achieving a 10,000x effective depth would require a starting raw coverage in the range of 100,000x.

To the reviewer's critical second point: even if 20,000x (or even 100,000x) raw coverage becomes affordable, it would not be a complete solution on its own. It only addresses one bottleneck: sequencing depth. The more fundamental limitation, as discussed in our commentary, is the absolute number of mutant DNA molecules in the initial blood sample. If a 10 mL blood draw from a lung cancer patient contains only five mutant DNA fragments for a specific variant, no amount of sequencing depth can overcome this sampling limitation. Therefore, ultra-deep sequencing is a necessary but not sufficient condition for ultra-sensitive ctDNA detection.

Regarding commercial tests, the landscape is varied. For MRD and screening applications, which require the highest sensitivity, platforms often aim for extremely high raw coverage. For instance, tests like Guardant Reveal or Signatera can reach effective (post-deduplication) depths well over 10,000x for specific hotspot regions. For therapy selection panels like Guardant360 CDx or FoundationOne Liquid CDx, the median raw coverage is typically in the range of 15,000-20,000x, which, after UMI deduplication, results in an effective coverage of approximately 1,500-2,500x. This allows them to reliably achieve a LoD of ~0.5%. The reviewer's mention of CancerSeek is interesting; however, it was a multi-analyte test combining protein markers with limited gene sequencing, and thus its coverage metrics are not directly comparable to comprehensive NGS profiling assays.

To address this last point in the article we added the following (p. 6, l. 13) :

“Major commercial therapy selection panels such as Guardant360 CDx or FoundationOne Liquid CDx typically achieve a raw coverage of ~15,000×, which after deduplication yields an effective depth of ~2,000×, consistent with their reported LoD of ~0.5%.”

  1. Lines 128-139: One cannot discuss tumor categories without adjusting for tumor volume. What is the minimal size of, say, lung or liver cancer lump, to be detectable by an “ideal” ctDNA assay from 10 ml of blood? I also do not understand the message of this section: if one considers cfDNA (normal and tumor DNA), why the increased concentration of cfDNA is better?! If this increased concentration is due to the excess of normal DNA, formal VAF value of 0.1% is no longer applicable, so even higher sensitivity of ctDNA detection is required. Please check this section.   

This is a crucial point, and we thank the reviewer for highlighting the ambiguity in this section. The original text was oversimplified. The reviewer is absolutely correct: a high total cfDNA concentration is only advantageous if it is accompanied by a sufficient absolute amount of ctDNA.

The minimal detectable tumor size is highly variable and depends on factors beyond volume, including tumor location, vascularization, proliferation rate, and the rate of apoptosis (ctDNA shedding rate). Studies have estimated that the lower limit of detection for many current assays corresponds to a tumor volume of approximately 1 cm³ (roughly 10⁸-10⁹ cells), but this is not a universal constant.

To clarify the central point of confusion: our argument is that tumors with high total cfDNA concentrations (like liver cancer) are often "high-shedders," meaning they release a greater absolute quantity of both normal and tumor DNA into the bloodstream. The key metric for detection is not the VAF (the fraction), but the absolute number of mutant genome equivalents (GEs) available for sequencing.

Let's clarify with an example:

  • Low-shedding tumor (e.g., Lung Cancer): 10 mL blood yields 5 ng/mL of plasma -> ~25 ng total cfDNA -> ~8,000 GEs. If the ctDNA fraction is 0.5%, we have only 40 mutant GEs in the entire sample. Detecting these is a major challenge.
  • High-shedding tumor (e.g., Liver Cancer): 10 mL blood yields 46 ng/mL of plasma -> ~230 ng total cfDNA -> ~76,000 GEs. Even if the ctDNA fraction were lower, say 0.2%, we would still have 152 mutant GEs—nearly four times as many target molecules, making detection far more robust.

The higher total cfDNA in the second case is beneficial because it correlates with a larger pool of available ctDNA molecules, despite the VAF potentially being low. We have rewritten this section to make this logic explicit by replacing the following paragraph (p. 7, l. 24):

“The quantity of cfDNA in cancer patients is highly variable and influenced by factors such as tumor type, stage, and volume. While cancer patients generally exhibit higher cfDNA levels than healthy individuals, this pattern differs across histologies. For instance, lung cancers can have low cfDNA levels (5.23 ± 6.4 ng/mL), while liver cancers often show much higher levels (46.0 ± 35.6 ng/mL) [20]. This is critically important because the ultimate constraint on sensitivity is the absolute number of mutant DNA fragments in a sample. For example, a 10 mL blood draw from a lung cancer patient might yield only ~8,000 haploid genome equivalents (GEs). If the ctDNA fraction is 0.1%, this provides a mere 8 mutant GEs for the entire analysis, making detection statistically improbable. Conversely, the same volume from a high-shedding liver cancer patient could provide ~80,000 GEs. Even with the same 0.1% ctDNA fraction, this yields 80 mutant GEs, providing a much stronger signal that is more amenable to detection by ultra-deep sequencing. Therefore, while a high background of normal DNA presents challenges, the higher total DNA quantity from high-shedding tumors is often advantageous because it is typically associated with a greater absolute number of targetable tumor molecules.”

  1. What are “variants known to be artifacts” in the bioinformatic “blocked list”? – please explain.

We thank the reviewer for requesting clarification on this important bioinformatic tool. A "blocked list" (or "blacklist") is a curated filter used in variant calling pipelines to automatically remove recurrent false positives that arise from technical, rather than biological, sources. These artifacts can mimic true low-frequency variants and are a major challenge when lowering the LoD.

To clarify this point, we have added the following in the manuscript (p. 10, l. 10)

“These artifacts are recurrent false positives arising from non-biological sources, and filtering them is crucial for achieving a low LoD. Common examples include: (i) systematic base-calling errors in specific sequence contexts (e.g., GGC motifs); (ii) reads misaligned from homologous pseudogenes; (iii) oxidative damage-induced mutations (e.g., G>T transversions); and (iv) true somatic variants originating from clonal hematopoiesis (CHIP) in blood cells, which are not representative of the tumor's genome.”

Round 2

Reviewer 1 Report

Comments and Suggestions for Authors

Authors changed manuscript as per recommended and now I endorsed for publication. Manuscript is looks nicer than before,

Thanks